# The Role of Virulence Proteins in Protection Conferred by *Bordetella pertussis* Outer Membrane Vesicle Vaccines

**DOI:** 10.3390/vaccines8030429

**Published:** 2020-07-30

**Authors:** René H. M. Raeven, Naomi van Vlies, Merijn L. M. Salverda, Larissa van der Maas, Joost P. Uittenbogaard, Tim H. E. Bindels, Jolanda Rigters, Lisa M. Verhagen, Sabine Kruijer, Elly van Riet, Bernard Metz, Arno A. J. van der Ark

**Affiliations:** Intravacc (Institute for Translational Vaccinology), Antonie van Leeuwenhoeklaan 9, 3721 MA Bilthoven, The Netherlands; naomi.van.vlies@intravacc.nl (N.v.V.); merijn.salverda@intravacc.nl (M.L.M.S.); larissa.van.der.maas@intravacc.nl (L.v.d.M.); joost.uittenbogaard@intravacc.nl (J.P.U.); tim.bindels@intravacc.nl (T.H.E.B.); jolanda.rigters@intravacc.nl (J.R.); lisa.verhagen@intravacc.nl (L.M.V.); sabine.kruijer@intravacc.nl (S.K.); elly.van.riet@intravacc.nl (E.v.R.); bernard.metz@intravacc.nl (B.M.); arno.van.der.ark@intravacc.nl (A.A.J.v.d.A.)

**Keywords:** *Bordetella pertussis*, virulence factors, pertussis vaccine, OMV, outer membrane vesicles, bvg, Th17, 2-dimensional gel electrophoresis, antibody profiling, intranasal mouse challenge model

## Abstract

The limited protective immunity induced by acellular pertussis vaccines demands development of novel vaccines that induce broader and longer-lived immunity. In this study, we investigated the protective capacity of outer membrane vesicle pertussis vaccines (omvPV) with different antigenic composition in mice to gain insight into which antigens contribute to protection. We showed that total depletion of virulence factors (*bvg*(-) mode) in omvPV led to diminished protection despite the presence of high antibody levels. Antibody profiling revealed overlap in humoral responses induced by vaccines in *bvg*(-) and *bvg*(+) mode, but the potentially protective responses in the *bvg*(+) vaccine were mainly directed against virulence-associated outer membrane proteins (virOMPs) such as BrkA and Vag8. However, deletion of either BrkA or Vag8 in our outer membrane vesicle vaccines did not affect the level of protection. In addition, the vaccine-induced immunity profile, which encompasses broad antibody and mixed T-helper 1, 2 and 17 responses, was not changed. We conclude that the presence of multiple virOMPs in omvPV is crucial for protection against *Bordetella pertussis*. This protective immunity does not depend on individual proteins, as their absence or low abundance can be compensated for by other virOMPs.

## 1. Introduction

Whooping cough, which is caused by the strictly human pathogen *B. pertussis*, is still an important cause of vaccine-preventable infant death worldwide. This disease continues to be a global public health concern even in countries with high vaccination coverage. In the 1990s, clinical pertussis reemerged in well vaccinated populations of countries that switched from whole cell pertussis vaccines (wcPV) to acellular pertussis vaccines (acPV). To date, it is recognized that the duration of acPV-induced protective immunity is limited [1], as its dominant T-helper (Th) 2 immunity seems to provide relatively short-lived protection [2]. Moreover, acPV-induced immunity is directed to only a limited number of antigens (1 to 5) and, therefore, is vulnerable for pathogen adaptation as demonstrated by the current worldwide rise in antigen-deficient strains, such as strains lacking pertactin (Prn) [3,4]. Booster vaccination strategies with acPV, including the recently introduced maternal vaccination [5], protect vaccinated individuals of all ages against disease but are overall not successful as current acPVs do not seem to protect against infection and transmission [6,7]. Therefore, people immunized with acPV are still able to transmit pertussis to susceptible individuals, which may explain the increase of clinical cases of pertussis in adults and the continuing high mortality and morbidity of infants too young to be vaccinated [8]. By contrast, wcPV limits colonization of *B. pertussis* in the upper airways, as demonstrated in baboons [6]. The current situation calls for improved or novel vaccines that induce a broader and longer-lasting protection to prevent *B. pertussis* infection and disease and in addition prevent transmission.

A pertussis vaccine based on outer membrane vesicles (omvPV) is considered a promising next-generation pertussis vaccine [9,10]. Outer membrane vesicles (OMVs) are blebs (Ø of 80–160 nm) formed under natural or deliberately inflicted stress conditions from the outer membrane of Gram-negative bacteria such as *B. pertussis* [11]. The bacteria can benefit from OMVs for various functions such as host-pathogen interaction [12] and immune modulation [13]. OMVs are currently considered as a blueprint for vaccine development against multiple pathogens, either as antigen carrier or adjuvant [14,15]. Advantages of OMV vaccines are that they contain a broad variety of antigens in their natural conformation, have the perfect size for uptake by most antigen presenting cells [16], and hold intrinsic adjuvants such as endotoxin, or lipo-polysaccharide (LPS), as well as other pathogen-associated molecular patterns [14,17]. Immunization with omvPV provides protection against *B. pertussis* colonization in the lungs of mice [18,19,20,21]. In contrast to the acPV-induced Th2 immunity, omvPV mainly induces systemic Th1/Th17 immunity, next to Th2 responses [19,20,22]. This is similar to the systemic T-helper immunity induced by *B. pertussis* infection [23] that seems to be required for long-lasting protection against infection [24]. Importantly, omvPV induces high antibody levels of all subclasses against multiple antigens in mice, as the omvPV contains a broad variety of immunogenic, and potentially protective, virulence-associated outer membrane proteins (virOMPs) [25].

The expression of virOMPs is regulated by the *bvg*A/S regulatory system of *B. pertussis* [26,27]. The *bvg*(+) mode enables production of virulence factors, including the virOMPs, while in the *bvg*(-) mode hardly any virulence factors are produced [28,29]. Metz and co-workers showed that the level of virOMPs in wcPV is positively associated with the level of protection against infection, as measured in the Kendrick test [28]. Among the virOMPs detected in omvPV, two autotransporters, BrkA and Vag8, were identified as most abundant [25]. Both are highly immunogenic in mice [19,20,25]. Anti-BrkA and anti-Vag8 antibodies have also been observed following natural *B. pertussis* infections in humans [30,31] and multiple groups have demonstrated that anti-BrkA antibodies are functional [32,33] and that vaccination with purified BrkA or Vag8 alone provides protection against a *B. pertussis* infection in mice [34,35,36,37,38].

These data indicate that omvPV-induced immunity may rely on the presence of virOMPs such as BrkA and Vag8. However, it is currently unknown to what extent these antigens contribute to the protective capacity. Therefore, this study aims to investigate whether or not virOMPs are crucial in the protection conferred by omvPV. To investigate the relationship between virOMPs and protective immunity, we immunized mice with omvPVs of different antigenic composition and demonstrated that these virOMPs influence omvPV-induced immunity profiles and protective capacity in the intranasal mouse challenge model.

## 2. Results

### 2.1. The Level of VirOMP in omvPV Is Positively Related to Protection against B. Pertussis

The presence of virulence proteins may be a prerequisite for a potent omvPV as many of these proteins have been described as immunogenic and protective. Since omvPV mainly contains virOMPs [25], we decided to focus on a possible relationship between the percentage of virOMP in the vaccine and the level of protection against *B. pertussis* infection. To investigate the role of virOMP, a series of omvPV varying in percentage of virOMP (ranging from high to low % virOMP: omvPV-wtB1917, omvPV-*bvg*(+)Toh and omvPV-*bvg*(-)Toh) was produced and tested next to two whole cell vaccines (wcPV-wtB1917, wcPV-Kh96/1) in the intranasal mouse challenge model (Figure 1).

Prior to the animal experiment, the proteome profiles of all vaccines under test were analyzed by LC-MS. Generally, of the approximately 3300 proteins in *B. pertussis*, ~700 could be quantified in wcPV and ~475 in omvPV made of virulent *B. pertussis*. In omvPV-*bvg*(-)Toh, approximately 530 proteins were detected. When dividing these identified proteins based on localization, it was observed that the level of OMPs in wcPVs and omvPVs was ~25% and ~60%, respectively (Figure 2A). The, omvPV-*bvg*(-)Toh contained only 5% OMPs and was enriched in periplasmic and other proteins. Subsequently, the localization of proteins was further specified for virulence proteins, based on a list of 141 virulence proteins described by Streefland et al. [39] of which 16 are considered virOMPs (Table 1). This analysis confirmed that the vast majority of virulence proteins in the omvPV-wtB1917 (~92% of %OMP) and omvPV-*bvg*(+)Toh (~92%) were virOMPs (Figure 2B). In both wcPV, ~30% of detected proteins were virulence factors of which ~67–83% virOMPs. The omvPV-*bvg*(-)Toh contained only some traces of virulence factors. The four virOMPs that were most abundant in omvPVs were BrkA, Prn, TcfA, and Vag8, although composition between the different omvPVs varied (Figure 2C). Surprisingly, omvPV-*bvg*(+)Toh contained a considerable amount of bipA (± 5%), an intermediate virulence protein that was not expected to be expressed during the logarithmic growth phase of *B. pertussis* [40]. FHA was very abundant in wcPVs but hardly present in omvPVs. The collective percentage of the 10 other virOMPs in all omvPVs and wcPVs was very low.

The level of protection induced by all vaccines under test was estimated in the intranasal mouse challenge model by calculating the Area under Curve (AuC) of the colonization curves, based on presence of colony forming units (CFU) during a time period of 7 days, in which a lower AuC corresponds to a better protection (Figure 3A, Appendix A). OmvPV made of virulent *B. pertussis* strains (wtB1917 and *bvg*(+)Tohama I) were clearly more protective than both wcPVs (wcPV-wtB1917 and wcPV-Kh96/1), while omvPV-*bvg*(-)Toh was hardly protective. When the percentage of virOMPs in these vaccines is plotted against their level of protection, a clear relationship can be observed where higher percentages of virOMPs provide better protection (Figure 3B). Overall, this observation suggests that the level of virOMPs in omvPV is positively related to the degree of protection in the lungs against a *B. pertussis* infection.

### 2.2. Immunoproteomic Profiling of High Antibody Responses Induced by omvPV-bvg(-)Toh and omvPV-bvg(+)Toh Reveals Partial Distinct Antigen Specificity

We observed clear distinct differences in antigen composition and protective capacity of omvPV-*bvg*(-)Toh and omvPV-*bvg*(+)Toh, raising questions to which antigens the antibody responses of both products were directed. To that end, antibody profiles in sera from both groups collected at day 59 (3 days post challenge) were compared using a Western blotting strategy. First, serum IgG responses against well-known purified acPV-antigens: filamentous hemagglutinin (FHA), fimbriae 2/3 (Fim2/3), pertactin (Prn), and pertussis toxin (Ptx) were checked. As expected, antibody responses against these virulence factors were not present after omvPV-*bvg*(-)Toh immunization, while only clear responses against Prn and FHA were observed after immunization with omvPV-*bvg*(+)Toh (Figure 4A). Responses against Fim2/3 and Ptx were low or absent as these proteins are also lowly abundant in omvPV.

Next, the sera were tested on a total B1917 whole cell lysate to investigate the antibody profile against all proteins present in virulent B1917. Here, a clear difference in antibody patterns was observed between mice immunized with omvPV-*bvg*(-)Toh or omvPV-*bvg*(+)Toh (Figure 4B). Especially in the higher molecular range, antibody binding, most likely against Vag8 and BrkA [19], was detected in omvPV-*bvg*(+)Toh sera, and this was absent in omvPV-*bvg*(-)Toh sera. We previously demonstrated that omvPV immunization evoked intense anti-LPS IgG3 antibodies [25]. Anti-LPS IgG3 antibody responses were also present in mice immunized with omvPV-*bvg*(-)Toh or omvPV-*bvg*(+)Toh at comparable levels, indicating that these are independent of *bvg*-status (Figure 4C).

Finally, 2D electrophoresis was combined with Western blotting (2DEWB) to identify in more depth the differences in serum antibody profiles between mice immunized with omvPV-*bvg*(-)Toh or omvPV-*bvg*(+)Toh. 2DEWB was performed in triplicate for each group (Appendix A). In total, 24 spots were detected of which the spot intensity, depicted as average gray value obtained from Delta2D, and identity, identified using LC-MS/MS, were summarized in a heatmap (Figure 4D and Appendix A). Depending on the antibody profiles, the detected immunogenic proteins could be divided into five clusters. Cluster I contained antibodies solely present in the omvPV-*bvg*(-)Toh group, which were directed against two cytosolic proteins, carB and the methyltransferase domain protein. Antibodies present in both groups but more abundant after omvPV-*bvg*(-)Toh immunization in cluster II included unidentified protein U1 and GroEL. Six non-virulent proteins in cluster III were found equally immunogenic in both groups that included GroEL, odhB, elongation factor Tu, and U2–4. Cluster IV contains antibodies against four proteins present in both groups but more abundant in the omvPV-*bvg*(+)Toh group. Antibodies against protein U5 were clearly present in both groups, whereas antibody responses against Vag8, rpsA and U6 were clearly present in mice immunized with omvPV-*bvg*(+)Toh and only slightly visible on blots of mice immunized with omvPV-*bvg*(-)Toh. Finally, cluster V contained 10 immunogenic proteins that evoked antibody responses only in the omvPV-*bvg*(+)Toh group. This cluster contained one unidentified protein (U7), six virulence factors (Prn, bteA and three different fractions of BrkA) and four non-virulent proteins (odhL, dadA, ahcY, aceF). Overall, these data reveal that both omvPV-*bvg*(+)Toh and omvPV-*bvg*(-)Toh induce strong antibody responses in mice that show some overlap in antibody profiles, yet particularly the ones against virulence factors were distinct in presence or magnitude, which may explain the difference in level of protection between both vaccines.

### 2.3. Deletion of Either Vag8 or BrkA in Outer Membrane Vesicles Does Not Affect Its Protective Capacity and Immunity Profiles

As BrkA and Vag8 were observed as most abundant and most immunogenic virOMPs in the omvPV (Figure 2C and Figure 4D), we decided to investigate the protective capacity of these antigens and the underlying immune responses induced by omvPV. To that end, a BrkA knockout mutant and a Vag8 knockout mutant of *B. pertussis* B1917 were constructed. The OMVs harvested from these mutants are here referred to as omvPV-∆BrkA and omvPV-∆Vag8. Proteome analysis indicated that both vaccines contained less virOMPs (12.8% and 28.1% less, respectively) and significantly more cytoplasmic proteins (13.3% and 20.0% more, respectively) compared to the omvPV-wtB1917 (Figure 2A,B). As expected, BrkA and Vag8 were absent in omvPV-∆BrkA and omvPV-∆Vag8, respectively (Figure 2C). The omvPV-∆BrkA contained mainly Vag8, TcfA and Prn whereas omvPV-∆Vag8 contained mainly BrkA, TcfA and Prn (Figure 2C).

### 2.4. Immunization with omvPV-∆BrkA and omvPV-∆Vag8 Reveals Equal Protective Capacity as omvPV-wtB1917

First, protection against colonization of *B. pertussis* conferred by omvPV-∆BrkA, omvPV-∆-Vag8 and omvPV-wtB1917 was assessed in the intranasal mouse challenge model. Saline (naive mice) and wcPV-Kh96/1 were included as negative and positive controls, respectively. The AuC of CFU present in the lungs, trachea and nasal cavity from day 3 to 7 post-challenge, was calculated. No differences in protective capacity were found between the different vaccinated groups, while all vaccinated groups showed a lower AuC compared to the naive mice in lungs, trachea and nose (Figure 5A). The kinetics of CFU present in the respiratory tract are depicted in Appendix A. Although no significant differences were observed between the vaccinated groups, it appeared that the omvPV-∆Vag8-immunized mice had a delayed nasal clearance from day 5 post-infection onwards. Notably, no complete clearance was achieved on day 7 post-infection in the nasal cavity of any experimental group. Whereas the percentage of virOMPs in the omvPV-∆BrkA (56%) and omvPV-∆Vag8 (38%) was lower compared to regular omvPV-wtB1917 (60%) (Figure 2B), the level of protection was not affected (AuC = 5), indicating that less than 40% virOMPs in omvPV-∆Vag8, at least with that specific antigen composition, was still effective (Figure 3B). These results provide support that protection provided by the omvPV-induced immunity is broader than only against either of its two most abundant immunogenic virulence factors as deletion of either Vag8 or BrkA in omvPV had no effect on the level of bacterial colonization throughout the respiratory tract.

### 2.5. Deletion of BrkA or Vag8 Has Limited Effect on Magnitude, Specificity and Subclass Distribution of Antibody Responses against Other Antigens

The lack of Vag8 or BrkA changed the antigen composition of the omvPV, potentially influencing the humoral immune responses. Therefore, we next determined potential differences in immunization-induced serum IgG (subclass) antibody levels against Ptx, Prn, FHA, Fim2/3, BrkA, Vag8, and OMV between the different groups 7 days (day 35, expected peak of plasma B-cells) and 28 days post booster immunization (day 56, moment before challenge). Results indicated significantly enhanced anti-OMV IgG responses in all immunized groups at day 35 (Appendix A) and day 56 (Figure 5B) as compared to naive mice. Immunization with omvPV-∆BrkA or omvPV-∆Vag8 resulted in absent anti-BrkA or anti-Vag8 IgG responses, respectively. Anti-BrkA as well as anti-Vag8 antibody responses were present in all other immunized groups. At day 56, immunization with omvPV-∆Vag8 induced a slight increase in anti-BrkA IgG compared to immunization with omvPV-wtB1917 or wcPV-Kh96/1, which may be the result of a higher BrkA presence in the omvPV-∆Vag8. Immunization with omvPV-∆BrkA and omvPV-∆Vag8 led to increased anti-Prn and anti-FHA responses on day 35 (Appendix A) and an increased anti-FHA response on day 56 (Appendix A) as compared to omvPV-wtB1917-immunized mice. Moreover, wcPV-Kh96/1 immunization induced significantly higher anti-FHA and anti-Ptx antibody responses as compared to all OMV immunized groups. These proteins were also more abundant in wcPV as compared to omvPV. Anti-Prn responses on day 56 were similar for all immunized groups whereas responses against Fim2/3 did not show clear results, since levels were highly variable.

IgG subclass responses measured on day 56 indicated that immunization with omvPV-wtB1917 resulted in anti-OMV antibody responses of all subclasses measured (Figure 5C). Both omvPV-∆BrkA and omvPV-∆Vag8-immunized mice had similarly broad, though slightly different subclass responses. Anti-OMV IgG1 and IgG2a responses were significantly reduced in the omvPV-∆Vag8-immunized mice. Notably, omvPV-∆BrkA-immunized mice contained slightly higher levels of anti-Vag8 IgG1 antibodies as compared to omvPV-wtB1917-immunized mice, whereas the IgG2a and IgG2b responses remained the same. Compared to immunization with OMV vaccines, wcPV-Kh96/1 immunization induced higher anti-OMV, anti-BrkA and anti-Vag8 IgG1 responses, while lower IgG2a, IgG2b and IgG3 responses were observed. Subclass analysis of the responses against Prn, Fim2/3 and Ptx induced by the three omvPVs revealed no large differences in subclass profiles (Appendix A). Notably, the anti-Prn antibody responses following immunization with all vaccines were mainly IgG1 responses, whereas the subclass distribution for all other antigens was more diverse. Anti-FHA responses after immunization with omvPV-wtB1917 showed an increased IgG3 response, while immunization with omvPV-∆BrkA and omvPV-∆Vag8 led to increased anti-FHA IgG1, IgG2a and IgG2b levels, although the relative concentrations of FHA in all three products were similar at 0.2%. Subclass responses upon wcPV-Kh96/1 immunization resulted mostly in IgG1 responses directed against Ptx, Prn, FHA, and Fim2/3. These observations indicate that the deletion of either Vag8 or BrkA hardly had any effect on the magnitude, specificity and subclass distribution of omvPV-induced antibody responses, with the exception of responses against the deleted antigen itself.

### 2.6. Number of omvPV-Induced Plasma and Memory B-Cells Are Not Influenced by Deletion of BrkA or Vag8

Antigen-specific IgG-producing plasma B-cells were measured in the spleen on day 35 (7 days post-booster immunization) (Figure 5D). Results indicated that immunization with omvPV-∆BrkA and omvPV-∆Vag8 resulted, just like regular omvPV-wtB1917, in significant induction of OMV-specific plasma B-cells as compared to naive mice. However, no difference in numbers between the types of OMVs was observed. As expected, BrkA-specific and Vag8-specific plasma B-cells were absent in mice immunized with omvPV-∆BrkA and omvPV-∆Vag8, respectively, but present in mice immunized with omvPV containing the respective proteins.

Antigen-specific IgG memory B-cells were determined in the spleen on day 56 (28 days post-booster immunization) (Figure 5E). Immunization with all three types of OMVs as well as the wcPV-Kh96/1 resulted in induction of OMV-specific memory B-cells as compared to naive mice but only a significant difference was seen for groups immunized with omvPV-∆BrkA or regular omvPV-wtB1917. Moreover, the number of OMV-specific memory B-cells was slightly lower in mice immunized with omvPV-ΔVag8. BrkA-specific and Vag8-specific memory B-cells were low or not detectable in all groups. Therefore, these observations support that the deletion of neither Vag8 nor BrkA affected the total numbers of antigen-specific IgG-producing plasma B-cells and IgG memory B-cells induced by omvPV.

### 2.7. The omvPV-Induced Mixed Th1/Th2/Th17 Response Is Not Affected by the Deletion of Either Vag8 or BrkA

To determine whether the absence of Vag8 or BrkA in omvPV impacts the magnitude and type of the systemic Th response, splenocytes isolated on day 56 were stimulated with either OMV, Vag8 or BrkA. Concentrations of Th1-related (IFNγ, TNFα), Th2-related (IL-4, IL-5, IL-13), Th17-related (IL-17A), and Treg/Th2-related (IL-10) cytokines were determined in the culture supernatants.

Immunization with all three types of omvPV resulted in significantly enhanced responses of all measured cytokines compared to naive mice, with the exception of TNFα, which was also strongly induced in naive mice upon stimulation with OMV (Figure 6A–C). Similarly, stimulation of splenocytes with Vag8 or BrkA significantly increased all cytokine responses in mice immunized with omvPV containing these respective proteins compared to naive mice, indicating the presence of Vag8- and BrkA-specific T-cells (Figure 6B,C). The absence of BrkA or Vag8 did lead to the absence of T-cell responses against these respective antigens but did not result in significantly altered cytokine levels when splenocytes were stimulated with OMVs.

Stimulation of splenocytes of wcPV-Kh96/1-immunized mice also demonstrated a mixed cytokine response (Figure 6A–C). Whereas stimulation with OMV indicated no large differences between mice immunized with omvPVs and those immunized with wcPV-Kh96/1, the cytokine production following BrkA and Vag8 stimulation was significantly lower in the wcPV-Kh96/1 group as compared to the omvPV-wtB1917 group, which is in agreement with the more abundant levels of Vag8 and BrkA in omvPV as compared to wcPV. These observations indicate that the mixed Th1/Th2/Th17 response induced by omvPV was not influenced by the deletion of either Vag8 or BrkA.

## 3. Discussion

The presence of a broad range of virulence factors in a pertussis vaccine is essential as protective immunity against these proteins assists in preventing modulation of host immune responses during infection [41]. Moreover, a broader range of immunity enables better protection against antigen-deficient strains such as circulating Prn-deficient strains [3,4]. Virulence factors are therefore potential vaccine targets. The antigen composition of omvPV and wcPV can be steered by choices in culture conditions (like medium [42,43], temperature [43], nutrients [44], and time of harvest), since these conditions as well as disturbances during production may affect the *bvg*-regulated expression of virOMPs in *B. pertussis* [45,46]. In the current study, we investigated the relationship between the level of virOMPs in omvPVs using detailed proteome analysis by LC-MS and the level of protection they confer in an intranasal mouse challenge model.

We demonstrated in this study that the presence of virOMPs in omvPV is positively correlated with the level of protection. The total level of virOMPs in omvPV seems, therefore, a good indicator for protection. Although less virOMPs were detected in omvPV-*bvg*(+)Toh (± 50%) and omvPV-∆Vag8 (± 40%) compared to omvPV-wtB1917 (± 70%), the clearance provided by omvPV-*bvg*(+)Toh and omvPV-∆Vag8 was as efficient as omvPV-wtB1917. This may indicate an optimum or threshold in required percentage of virOMPs in omvPV to induce maximum protection or, alternatively, the virOMP composition, that varies between the studied omvPVs, may play a role. It is not clear how much specific virOMPs contribute to protection against *B. pertussis* infection individually. Vag8, BrkA, TcfA, and Prn are the most abundant and immunogenic virOMPs present in omvPV [25]. Gasperini et al. previously detected some of these virOMPs (BrkA, TcfA and Vag8) but also BipA and SphB1 as adhesins in omvPV-*bvg*(+) and demonstrated that these are potentially protective in purified form [37].

The antigenic composition of omvPV depends on multiple parameters, as its production includes cultivation of *B. pertussis* whole cells and an OMV extraction method [47]. The strain used, in combination with the culture conditions of *B. pertussis*, determines the quality of the whole cell harvest and, therefore, the composition of the omvPV. To obtain a highly potent whole cell harvest, the *bvg*A/S molecular switch [26,27] should be in the *bvg*(+) mode, assuring maximal expression of virulence factors. In the proteome analysis we noticed that choice of strain and culture medium affected levels and composition of virOMPs in omvPV. The presence of virOMPs may be higher in B1917, a post-vaccination era *ptx*-promotor 3 (P3) strain, as compared to pre-vaccination era *ptx*-promotor 1 (P1) strains 509, 134 (both present in wcPV-Kh96/1) and Tohama I, as P3-strains are identified as more virulent [48]. Culturing *B. pertussis* strain B1917 in THIJS medium results in a high percentage of OMP in omvPV that are almost all virOMPs. Culturing Tohama I in a non-defined Verwey medium also resulted in high expression of virOMPs in omvPVs, although this was lower than observed on omvPV-wtB1917 and a bigger fraction of the OMP was non-virOMP (± 5%). For the Tohama I mutants, omvPV-*bvg*(+)Toh expressed approximately 5% BipA that was not observed in products derived from B1917. BipA (*bvg*-intermediate phase protein A) is involved in the transition from *bvg*(-) to *bvg*(+) during the intermediate phase (*bvg*(i)) [49]. However, bipA expression may also vary between *B. pertussis* strains in general [40]. For now, it is not clear when and to what extent culture conditions and/or genetic modifications can be used to affect *bvg*-regulated virulence and consequently the antigenic composition of omvPV in a controlled manner. More research is, therefore, required. Nevertheless, the percentage of virOMP of all vaccines under test concurred highly with the AuC of lung clearance in mice.

To improve the immunity of future pertussis vaccines, certain aspects of the immunity profiles provided by wcPV and *B. pertussis* infection are often used as blueprints [23,50]. These include systemic Th1/Th17 responses as well an antibody response against a broad range of antigens [25,51,52,53]. We and others showed before that omvPVs also provide these broad antibody responses [25] as well as mixed Th1/Th17 next to Th2 responses [19,20,22], which is in sheer contrast to the Th2-dominated acPV-induced response [54], which is directed to only a few antigens. In the current study, these observations were confirmed in a comparison with wcPV-Kh96/1. Nevertheless, some differences in magnitude and specificity of both antibody and T-cell responses were observed between omvPV-wtB1917 and wcPV-Kh96/1 as demonstrated by the lower Vag8-specific and BrkA-specific antibody levels and T-cells responses in the mice immunized with the wcPV. Moreover, whereas both wcPV and omvPV induced antibodies of all subclasses, the omvPV induced more IgG2a and IgG2b responses while the wcPV induced more IgG1 responses. Some antigens seem to be more IgG1-dominated, such as was observed in this study for Prn, even when present in omvPV and wcPV. This broad subclass response induced by omvPV and wcPV, in contrast to the IgG1-dominated acPV response [25], might provide a more comprehensive functionality of antibodies such as prevention of bacterial adherence and increased bactericidal activity and opsonophagocytosis [31,55,56], therefore, hampering the ability of *B. pertussis* colonization.

In this study, we demonstrated that raising immunity against virulence proteins is very effective as the absence of antibody responses against virulence proteins in omvPV-*bvg*(-)Toh resulted in an astonishing drop in the protective capacity. Mice immunized with these omvPV-*bvg*(-)Toh still produced high levels of antibodies that were directed against factors that were not influenced by the *bvg*(-) status, such as unidentified proteins (U1-5) (presumably lipoproteins, based on their location and behavior on the blots), LPS and GroEL, indicating that not only virOMPs in omvPV are immunogenic. Both LPS and GroEL have previously been described as protective antigens [32,57]. However, the omvPV-*bvg*(-)Toh-induced antibody responses were less effective since they hardly provided protection against infection. Overall, omvPV, which contain multiple virulence factors, evoke much broader antibody response when compared to acPV [25] and likely offer a broader base of protection against antigen-deficient strains. Zurita et al. indeed demonstrated that omvPV provided similar protection against Prn-expressing and Prn-deficient strains [58].

BrkA and Vag8 are two highly immunogenic virulence proteins that are abundant in omvPV [25]. These proteins are involved in complement evasion [59,60,61,62] and, therefore, are potential targets for improving pertussis immunity. Multiple groups have reported that immunization with purified or recombinant Vag8 or BrkA provides antibody responses and protection in the intranasal mouse challenge model [33,34,35,36,37,38,63]. Some suggest to use these antigens to replace or supplement current antigens in the acPV. Whereas this will most likely provide broader protection, this strategy will probably not change the Th2-dominated response induced by acPV. On the contrary, we have demonstrated previously [20,25,64] and in the current study that immunization with omvPV evokes broad antibody subclass responses and Th1/Th17 responses against BrkA and Vag8, among other antigens. Importantly, the lack of BrkA or Vag8 in omvPV did not result in reduced protective capacity against a *B. pertussis* infection as compared to the omvPV-wtB1917, although some small differences in immune responses were observed. In addition, these data indicated that deletion of the BrkA or Vag8 antigen does result in a shift in proteins present in omvPV, however, these proteins seem to compensate for the loss of BrkA or Vag8 as the change in composition does not greatly influence the magnitude and subclass profile of the humoral immunogenicity, nor the type of T helper cell responses induced by omvPV.

In conclusion, this study has demonstrated that omvPV-induced protection is heavily dependent on the presence of virOMPs, similar to what has been reported for wcPV before. However, our data demonstrated that although the strength of omvPV-induced immunity relies on the presence of virOMPs, this is not solely depending on either of the two most abundant immunogenic antigens (BrkA or Vag8). Moreover, the presentation of these virOMPs incorporated in omvPV in their natural conformation and with the right intrinsic adjuvants resulted in a broad antibody response and mixed T-helper 1, 2 and 17 responses, making omvPV a very effective vaccine candidate.

## 4. Materials and Methods

### 4.1. Bacterial Strains and Growth Conditions

*Escherichia coli* (E.c.) strains JM109 (Promega, Madison, WI, USA), TOP10F’ (Invitrogen, Carlsbad, CA, USA), and SM10 λpir [65] were grown in LB broth or on LB plates supplemented with suitable antibiotics. LB broth was prepared from LB medium capsules (MP biomedicals, Irvine, CA, USA), according to the manufacturer’s instructions. LB plates were prepared by supplementing LB broth with bacto-agar (BD). Plates and liquid cultures were grown overnight at 37 °C, liquid cultures were shaken at 200 RPM.

*Bordetella pertussis* (B.p.) strain B1917 was grown on plates prepared from Difco Bordet Gengou Medium Base (BD) supplemented with horse blood (BioTRADING, Mijdrecht, Utrecht, The Netherlands) and suitable antibiotics. Bordet Gengou horse blood (BG-HB) plates were incubated for 3–5 days at 35 °C and 5% CO_2_. Liquid cultures were grown overnight in THIJS broth [66] at 35 °C and 200 RPM.

### 4.2. Construction of BrkA and Vag8 Knockout Mutants

For deletion of the coding regions of BrkA and Vag8 in the B.p. B1917 genome, counter selectable suicide vector pSS1129 [67] was used. Details of counterselection-based genomic mutagenesis can be found in [67,68].

Constructs for the deletion of BrkA and Vag8 were created by fusion of the ~500 bp regions upstream and downstream of the genes by overlap-extension PCR [69]. First, the regions upstream (PCR Ia) and downstream (PCR Ib) of BrkA/ Vag8 were amplified by PCR using B.p. B1917 genomic DNA as template and using the primers shown in Table 2 (primer design was based on the B.p. B1917 genome sequence, GenBank: CP009751). For both BrkA and Vag8, PCR products Ia and Ib partially overlap in sequence, as a result of the non-annealing homologous parts in the long primers of both PCRs (shown in capital letters in Table 2). Both PCR products were mixed in a 1:1 ratio and then served as a template for a second PCR reaction (PCR II) using the primers indicated in Table 2. In PCR II, products Ia and Ib initially anneal on each other, which eventually results in fusion of the two PCR products. The resulting constructs roughly match the ~500 bp region upstream of BrkA/Vag8 fused to the ~500 bp region downstream of BrkA/Vag8.

Constructs were ligated in pGEM-T Easy vector (Promega) and then transformed in E.c. JM109 (Promega) according to the manufacturer’s instructions. Plasmid was then harvested from an overnight culture of a successful transformant in LB-Amp using the Wizard SV miniprep kit (Promega). Isolated plasmid was digested with EcoRI (New England Biolabs, Ipswich, MA, USA) and the digestion mix was separated by gel electrophoresis, after which the band containing the construct was sliced out and purified using the Wizard SV PCR clean-up kit (Promega). Constructs were ligated in pSS1129 digested with EcoRI and rSAP (New England Biolabs) after which the ligation mix was transformed in E.c. TOP10F’ (Invitrogen Waltham, MA, USA) according to the manufacturer’s instructions. Plasmid was harvested from an overnight culture of a successful transformant as described before and transformed in E.c. SM10 λpir by heat-shock at 42 °C for 2 min. Successful transformants were grown overnight and stored as glycerol stock until further use.

Plasmid pSS1129 with ΔBrkA or ΔVag8 construct was transferred from E.c. SM10 λpir to B.p. B1917 (Nal^R^, Strep^R^) by conjugation. Cells were harvested from plates and mixed on a BG-HB plate supplemented with 10 mM MgCl_2_. After incubation for 6 h at 35 °C and 5% CO_2_, the cell mixture was transferred to BG-HB plates supplemented with 50 µg/ mL nalidixic acid (to select against E.c. SM10) and 10 µg/ mL gentamycin (to select for cells that had integrated the plasmid into their genome as a result of homologous recombination; note that only such genomic recombinants can survive, because pSS1129 is unable to replicate in B.p. B1917).

Successful recombinants were then plated on BG-HB plates with streptomycin to select for spontaneous streptomycin resistant clones that result from a second crossover event between the homologous sequences that flank the integrated pSS1129. As this can result either in successful removal of BrkA/Vag8 or return to wildtype, recombinants were screened by PCR to identify successful knockouts (using the primers indicated in Table 2).

### 4.3. Whole Cell and Outer Membrane Vesicle Preparations and Analysis

In this study, the *B. pertussis* Dutch clinical isolate B1917 and *B. pertussis* mutants Tohama I *bvg*(-), Tohama I *bvg*(+) (kind gift of Marjolein van Gent, RIVM, Netherlands), B1917-ΔVag8, and B1917-ΔBrkA (Intravacc, Bilthoven, Netherlands) were used to prepare wcPV and/or omvPV. *B. pertussis* strains were grown in shaker flask cultures (200 rpm at 35 °C) containing 200 mL THIJS medium [66] or Verwey medium [70]. Whole cell suspensions were heat-inactivated for 30 min at 56 °C and OMV were extracted as described by Zollinger et al. [47]. The wild type *B. pertussis* B1917, B1917-ΔVag8 and B1917-ΔBrkA strains were cultured in THIJS medium and harvested mid logarithmic growth phase to warrant production of potent omvPV-wtB1917, wcPV-wtB1917, omvPV-ΔVag8, and omvPV-ΔBrkA [46]. OmvPV were prepared from Tohama I mutants of which the *bvg*-system was genetically locked in *bvg*(+) or *bvg*(-) mode (the resulting OMVs are referred to as omvPV-*bvg*(+)Toh and omvPV-*bvg*(-)Toh, respectively). Both Tohama I mutants were cultured in Verwey medium as the mutants did not grow well in THIJS medium. The lyophilized, non-adjuvated wcPV-Kh96/1 reference vaccine was included that contains whole cells of strain 509 (8 IOU/mL) and 134 (8 IOU/mL), cultured in B2 medium [43].

Total protein concentration was measured with the Peterson’s modification of the Lowry protein assay or BCA. Both assays were performed according to manufacturer’s protocol (Sigma-Aldrich, Zwijndrecht, The Netherlands and Thermo Fisher, Waltham, MA, USA respectively).

### 4.4. LC–MS/MS Analysis for Antigen Composition Vaccines

#### 4.4.1. Protein Digestion

Samples (omvPV or wcPV) were diluted in a denaturation buffer containing 1 M guanidine hydrochloride (Gnd-HCl) and 50 mM triethylammonium bicarbonate, pH 8.5, to a final protein concentration of 0.2 mg/mL. Reduction and alkylation of disulfide bridges was performed with 5 mM TCEP (Thermo, Rockford, IL, USA) (1 h at 55 °C) and 9.4 mM iodoacetamide (30 min at RT in the dark). Proteins were digested with 0.5 µg endoproteinase Lys-C (Roche, Mannheim, Germany) followed by incubation (4 h, 37 °C). Subsequently, digests were incubated (ON, 37 °C) with 1 μg trypsin (Promega, Madison, WI, USA). Solid-phase extraction was performed to remove excess reagents using C18 Sep-pack cartridges (Waters, Milford, MA, USA) according to the manufacturer’s protocol. Peptides were dried in a vacuum concentrator. Peptides were dissolved in 100 µL formic acid/DMSO/water (0.1/5/94.9% *v/v*) for LC–MS/MS analysis.

#### 4.4.2. Peptide Identification

Samples were analyzed by nanoscale reversed-phase liquid chromatography electrospray mass spectrometry, according to the method by Meiring et al. [71]. The analysis was performed on LTQ-Orbitrap XL mass spectrometer (Thermo Fisher Scientific, Bremen, Germany). Analytes were loaded on a trapping column (Reprosil-Pur C18-AQ 5 μm (Dr. Maish, Ammerbruch, Germany); 23 mm long × 100 μm inner diameter) with solvent A (0.1% (*v/v*) formic acid in water) in 10 min at 5 μL/min. The analytes were separated by reversed-phase chromatography on an analytical column (Reprosil-Pur C18-AQ 3 μm (Dr. Maish, Ammerbruch, Germany); 36.2 cm long × 50 μm inner diameter) at a flow rate of 100–150 nL/min. A gradient was started with solvent B (0.1% (*v/v*) formic acid in acetonitrile): 6% to 28% in 130 min, 28% to 38% in 10 min and 90% for 10 min. After the gradient, the columns were equilibrated in 100% solvent A for 20 min at 100–150 nL/min. The peptides were measured by data-dependent scanning; comprising a MS-scan (m/z 300–1500) in the orbitrap with a resolution of 60,000 (fwhm), followed by collision-induced dissociation (LTQ) of the 10 most abundant ions of the MS spectrum. The threshold value for these precursor ions was set at 1000 counts. The normalized collision energy was set at 35% and isolation width at 2.0 Da, activation Q to 0.250 and activation time to 30 ms. The maximum ion time (dwell time) for MS scans was set to 250 ms and for MS/MS scans to 1000 ms. Precursor ions with unknown and +1 charge states were excluded for MS/MS analysis. Dynamic exclusion was enabled (exclusion list with 500 entries) with repeat set to 1 and an exclusion duration of 15 s. The background ion at 391.28428 was used as lock mass for internal calibration.

#### 4.4.3. Data Handling for Protein Digests

Proteome Discoverer software (version 2.1, Thermo) was used for peak area determination, identification and relative quantification of the LC–MS/MS raw data. Identification of peptides was performed by searching MS/MS spectra against the protein database of *B. pertussis* Tohama (NCBI 257313) (3258 entries) using the SEQUEST HT mode. Asparagine deamidation and methionine oxidation were set as variable modifications and carboxymethylation of cysteine as a fixed modification. The data were searched with full trypsin cleavage specificity, allowing two miscleavages. Precursor ion and MS/MS tolerances were set to 5 ppm and 0.6 Da, respectively. Peptides were filtered to 1% FDR using Perculator (Proteome Discoverer, Thermo). The molar concentration of proteins was estimated according to Silva et al. [72]. The molar concentrations were converted to mass concentrations by multiplying with the molecular masses of the proteins. The percentage of protein abundance for each individual protein relative to the total sum of all identified proteins was used to calculate the relative protein composition. Cellular localization of identified *B. pertussis* proteins was based on prediction by PSORTb v3.0.2 and virulence factor content was based on the publication of Streefland et al. [39].

### 4.5. Ethics Statement

The welfare of the animals was maintained in accordance with the general principles governing the use of animals in experiments of the European Communities (Directive 2010/63/EU) and Dutch legislation (BWBR0003081). The first experiment was approved by the Ethical Committee on Animal experiments on 9 October 2014. The second experiment received approval (project # AVD3260020174285) by the Central Committee for Animal Studies on 15 January 2018 and approval by the Authority for Animal Welfare on 25 October 2018.

### 4.6. Immunization and Challenge of Mice

For the first experiment, the protective capacity of the selected wcPV and omvPV was assessed in a series of three experiments. Here, female, 8-week old BALB/c mice (Harlan, Horst, The Netherlands) were vaccinated twice (day 0 and day 28) with either 5 IOU wcPV-Kh96/1, 5 IOU wcPV-wtB1917, 4 µg of omvPV-wtB1917, omvPV-*bvg*(+)Toh and omvPV-*bvg*(-)Toh or saline (naive). Subsequently, all animals were challenged intranasally with 20 µL *B. pertussis* B1917 (2 × 10^5^ CFU) on day 56 followed by collection of lungs at day 59, 60, 61, 62, and 63 for analysis of CFU. In addition, serum was collected on day 59 for analysis of immunoproteomic profiling.

In the second experiment, mice were vaccinated twice (day 0 and day 28) with 4 µg of either omvPV, omvPV-ΔBrkA or omvPV-ΔVag8. As negative and positive control, mice were immunized with saline or 5 IOU/mL of the wcPV-Kh96/1, respectively. Subgroups of animals were euthanized on day 35 (analysis of antibody levels and B-cells) and day 56 (analysis of antibody levels, memory B-cells and T-helper-related cytokine production). On day 56, all remaining animals were challenged intranasally with 20 µL *B. pertussis* B1917 (2 × 10^5^ CFU), and subgroups of animals were euthanized on days 59, 61, and 63 to analyze the numbers of bacterial colonies in lungs, trachea and nose.

### 4.7. Sample Collection

Mice were anesthetized (isoflurane/oxygen) and sacrificed for immunological assays at day 35 for analysis of plasma cells and antibody responses (*n* = 8 per group). At day 56, mice were sacrificed for analysis of antibody, memory B-cell and T-helper responses (*n* = 8 per group, *n* = 5 for reference group). Serum for antibody detection was obtained by collecting whole blood in a serum collection tube (MiniCollect 0.8 mL Z Serum Sep GOLD, Greiner Bio-one, Alphen aan den Rijn, The Netherlands). After coagulation (30 min, RT), sera were collected by centrifugation (10 min, 3000 g) and stored at −80 °C. For B- and/or T-cell assays, complete spleens were collected in 5 mL RPMI complete medium (RPMI-1640 medium (Gibco, Waltham, MA, USA) supplemented with 10% FCS (Gibco), 100 units penicillin, 100 units streptomycin, and 2.92 mg/mL L-glutamine (Gibco)) and homogenized using a 70 μm cell strainer (BD Falcon, BD Biosciences, San Jose, CA, USA) by using a previously described protocol [20]. Erythrocytes in spleen samples were lysed by treatment with Ammonium-Chloride-Potassium (ACK) lysis buffer (Gibco, Waltham, MA, USA). For colonization assays, isolated lungs and trachea of mice (*n* = 5 per group, per time point) were homogenized in 2.4 mL THIJS medium [66] using the GentleMACS Octo Dissociator (Miltenyi Biotec, Bergisch Gladback, Germany), while nasal lavages were obtained by flushing the nose with 1 mL THIJS medium supplemented with 40 μg/mL cephalexin.

### 4.8. Colonization Assays

Lung and trachea homogenates, and nose lavages were serially diluted (undiluted, 1:10, 1:100, and 1:1000 depending on organ type) in THIJS medium containing 1% THIJS supplement. For nasal lavages, THIJS medium with 1% THIJS supplement was also supplemented with Cephalexin (4 mg/mL, Sigma, Zwijndrecht, The Netherlands). For each sample, 100 µL was plated on Bordet–Gengou agar plates with 15% sheep blood (BD, Vianen, The Netherlands), evenly distributed on the plates by using glass beads and incubated for 5 days at 35 °C. The number of CFU/mL was determined using a colony counter (ProtoCOL, Synbiosis, Cambridge, UK). A detection range for this method between 50–500 CFU/mL was used.

### 4.9. Multiplex Immunoassay (MIA) for Antibody Measurements

Antibodies against *B. pertussis* antigens mutant Ptx, Prn, FHA, Fim2/3, BrkA, Vag8, and OMV were measured using a MIA. Conjugation of antigens and OMVs to magnetic beads of different bead regions (Luminex, Austin, TX, USA) was performed using the Bio-Plex Amine Coupling kit (Bio-Rad, Hercules, CA, USA) and was described previously [20]. Serum was diluted 1:100 for IgG (subclass) and 1:1000 for anti-OMV IgG in PBS containing 0.1% Tween 20 and 3% bovine serum albumin (Sigma, Zwijndrecht, The Netherlands) and added to the conjugated beads in a 1:1 ratio. Subsequently, samples were incubated with *R*-Phycoerythrin (RPE)-conjugated anti-mouse IgG (1:200), IgG1 (1:200), IgG2a (1:40), IgG2b (1:200), and IgG3 (1:200) (Southern Biotech, Birmingham, AL, USA). Data was acquired with a Bio-Plex 200, processed using Bio-Plex Manager software (v6.1, Bio-Rad Laboratories, Hercules, CA, USA), corrected for the background signal, and presented as fluorescence intensities (FI). The limit of detection of each analyte was set at 15% of the highest FI-value in the dataset.

### 4.10. B-cell ELISpot for Plasma and Memory B-cells

For the detection of OMV, BrkA or Vag8-specific IgG-producing plasma cells in spleen, filter plates were coated with 10 µg/mL WT B1917 OMV, 5 µg/mL BrkA or 5 µg/mL Vag8 and the ELISpot method was used as described before [20]. Spots were counted with an AID iSpot reader (Autoimmun Diagnostika, Strassberg, Germany) and indicated as antibody-secreting cells (ASC) per 5 × 10^5^ cells. For analysis of memory B-cells, splenocytes were stimulated to induce differentiation into antibody-secreting cells as described previously [21]. The amounts of OMV, BrkA and Vag8-specific ASC were subsequently determined by using the same ELISpot method, only here 1 × 10^5^ stimulated cells were added to the coated plates.

### 4.11. Cell Stimulation and MIA for T-Helper (Th) Cytokine Analysis

Splenocytes (1 × 10^6^ cells) were stimulated for 3 days with 1.5 μg/mL WT B1917 OMV, 1 μg/mL BrkA or 1 μg/mL Vag8 using a method that was described before [20] to induce cytokine production after which the supernatant was collected. The concentration of T-helper subset cytokines interleukin-4 (IL-4), IL-5, IL-10, IL-13, IL-17A, TNFα, and IFNγ was determined in the supernatants using a ProcartaPlex Mix&Match Mouse 7-Plex (ThermoFisher, Waltham, MA, USA). The supernatants were diluted twice (for IL-4, IL-5, IL-10, IL-13 and TNFα) or diluted 10 times (for IFNγ and IL-17A) before analysis. Data were acquired with a Bio-Plex 200 (Bio-Rad, Hercules, CA, USA) and analyzed using Bio-Plex Manager software (v6.1, Bio-Rad). The background of the RPMI complete medium control per mouse per cytokine was subtracted from the results, and the results were calculated in pg/mL.

### 4.12. Immunoproteomic Antibody Profiling

The method for SDS PAGE, 2D electrophoresis and Western blotting (2DEWB), Delta2D analysis, in-gel-digestion and LC-MS/MS was described previously [25]. Briefly, for SDS PAGE, 1 µg single antigen (FHA, Fim2/3, Ptx, Prn) or 10 µg B1917 lysate was separated on gel. Proteins were either stained with Imperial protein stain or proteins were transferred to a nitrocellulose membrane and blots were blocked overnight. Subsequently, IgG blots and IgG3 blots were incubated with 1:1000 pooled serum taken on day 59 from mice immunized with omvPV-*bvg*(-)Toh or omvPV-*bvg*(+)Toh, followed by incubation with IR800-labelled goat-anti-mouse IgG or IgG3 antibodies. Additionally, 1:200 anti-LPS (Mab 88F3) was used as control for anti-LPS IgG3 antibody responses. Odyssey infrared imager was used for visualization of the immunogenic proteins.

For 2DEWB, 25 µg B1917 was rehydrated on an IEF strip pI 3–10NL overnight. IEF separation was performed followed by a double equilibration of the strip using equilibration buffer containing DTT or iodoacetamide respectively. Proteins were again either stained with Imperial protein stain or proteins were transferred to a nitrocellulose membrane. Blot was blocked overnight, followed by an incubation with 1:100 diluted pooled serum taken on day 59 from mice immunized with omvPV-*bvg*(-)Toh or omvPV-*bvg*(+)Toh. Subsequently, blots were incubated with 1:5000 IR800-labelled goat-anti-mouse, and immunogenic proteins were visualized using the Odyssey infrared imager. Each 2DEWB was performed in triplicate.

Delta2D analysis was performed to analyze the 2DEWB. Triplicate of 2DEWB blots of both groups were warped using group strategy. Average gray values were used for spot intensities. In gel-digestion and subsequent LC-MS/MS analysis was used for protein identification of immunogenic proteins. To generate the heatmap, the Delta2D data and in-gel-digestion data were combined. Two filters were applied on the data for the clustering. (i) The average gray value of the triplicates had to be ≥0.01 and the signal should be present on two blots within a group. (ii) If the signal in both groups is ≥0.01, a fold change of ≥2 was applied to determine if the signal was increased in one group compared to the other group.

### 4.13. Statistics

The total number of mice for the assessment of lung colonization (*n* = 5) is based on a power calculation (difference ≥ 0.9 log10, power = 80%) using datasets from previous experiments. For the analysis of immunological parameters such as B-cell and T-cell responses, eight mice per group allowed a reliable analysis.

To determine whether there were differences in lung colonization after challenge between the groups, taking into account both intensity of colonization as well as duration, the AuC was calculated for each animal and possible differences between the means of the groups were analyzed with one-way ANOVA. *p*-values were corrected for multiple comparisons using the False Discovery Rate method of Benjamini and Hochberg.

Before statistical analysis, antibody data and the results from the T- and B-cell measurements were log-transformed. Differences in antibody levels, splenocyte cytokine levels, and B-cell numbers between groups were first tested for normality and subsequently analyzed using the Kruskal–Wallis test. *p*-values were corrected for multiple comparisons using the Dunn’s multiple comparison test. As a change in splenocyte cytokine level of less than 1.5-fold compared to saline-vaccinated (naive) mice is considered not to be biologically relevant, no statistical analyses were performed on the data of groups which did not show more than a 1.5-fold change.

Differences between groups were considered significant when *p* < 0.05. Statistical analyses were performed using GraphPad Prism version 8.1.2.

## 5. Conclusions

Our data demonstrates that the level of virOMPs present in omvPV is crucial for induction of protection against *Bordetella pertussis*. As omvPV contain virOMPs in their natural conformation as well as intrinsic adjuvants, omvPV induces a broad antibody response and mixed T-helper 1, 2 and 17 responses. The level of protection and the type of immunity provided by omvPV does not seem to depend on individual proteins, as their absence or low abundance can be compensated for by other virOMPs as we showed for the two most abundant immunogenic antigens (BrkA or Vag8). Overall, these data demonstrate the potential of omvPV as very effective vaccine candidate against *Bordetella pertussis*.

## Figures and Tables

**Figure 1 vaccines-08-00429-f001:**
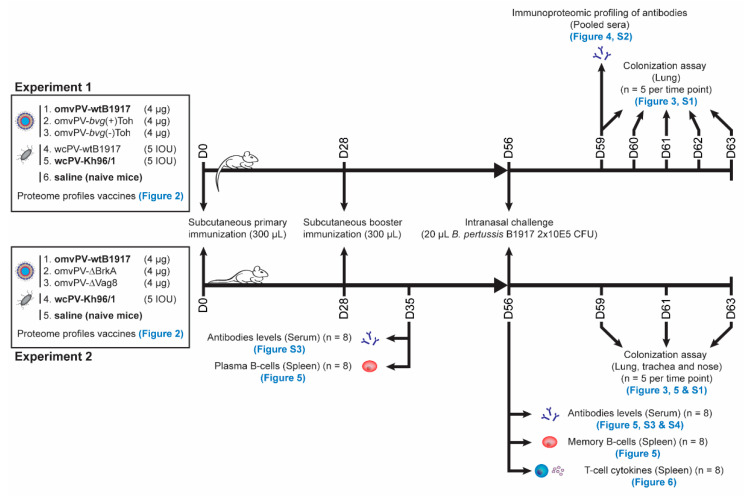
Experimental study design. The used vaccines, vaccine dosage, immunization schedule, strain and dosage of challenge culture, and days of sample collection are depicted for both experiments. The vaccines in bold were used in both experiments. For each time point, the experimental action or analysis is mentioned including the number of included animals and references to the figures in this manuscript where data is depicted.

**Figure 2 vaccines-08-00429-f002:**
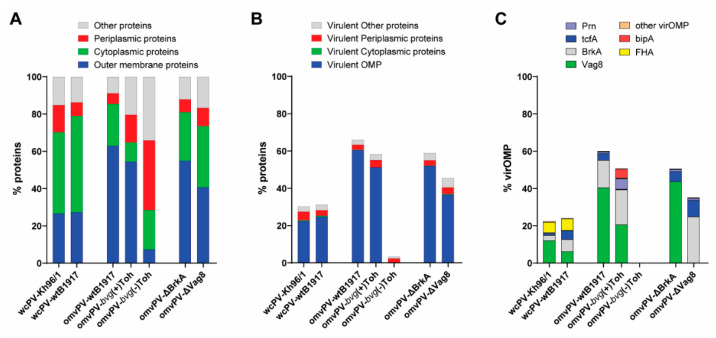
Proteome profiles of whole-cell pertussis vaccine (wcPV) and outer membrane vesicle pertussis vaccine (omvPV). (**A**) Proteins detected with LC-MS in wcPV and omvPV used in this study were divided in mass fractions (%) based on cellular location such as the outer membrane, periplasm, cytoplasm, and other proteins. (**B**) The total percentage of virulent proteins was divided in mass fractions (%) based on cellular location as described above. (**C**) The total percentage virulence outer membrane proteins (virOMP) was further divided in the percentage of six individual virOMPs (bipA, BrkA, FHA, Prn, TcfA, Vag8) next to the sum of other virOMPs.

**Figure 3 vaccines-08-00429-f003:**
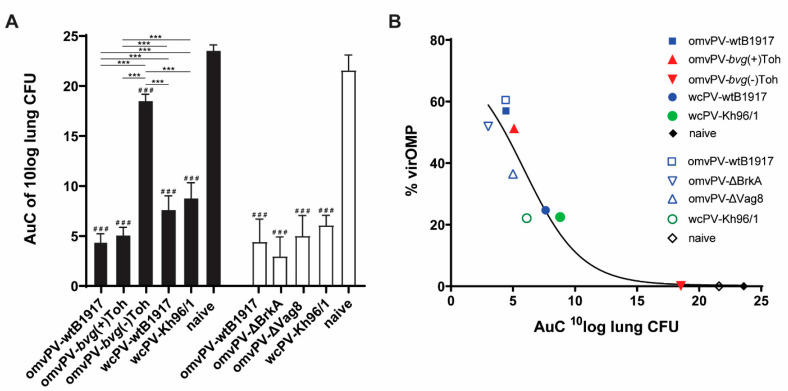
Relation between percentage of virulence outer membrane proteins (virOMP) and area under the curve (AuC) of lung colonization in mice. (**A**) The level of protection of all vaccines under test in this study, divided over experiment 1 (black bars) and 2 (white bars), expressed as AuC of ^10^log lung colony forming units (CFU) during 7 days after intranasal challenge of mice with virulent *B. pertussis* strain B1917. # is significant difference compared to naive group. Significant differences compared to the naive mice are indicated as ^#^
*p* ≤ 0.05, ^##^
*p* ≤ 0.01 and ^###^
*p* ≤ 0.001. Significant differences between experimental groups is depicted with a line between both groups and * *p* ≤ 0.05, ** *p* ≤ 0.01 and *** *p* ≤ 0.001. (**B**) The % virOMP in all tested vaccines plotted against the AuC of ^10^log lung colonization as a relation between the % virOMP and AuC of lung colonization in mice using a sigmoidal curve fitting.

**Figure 4 vaccines-08-00429-f004:**
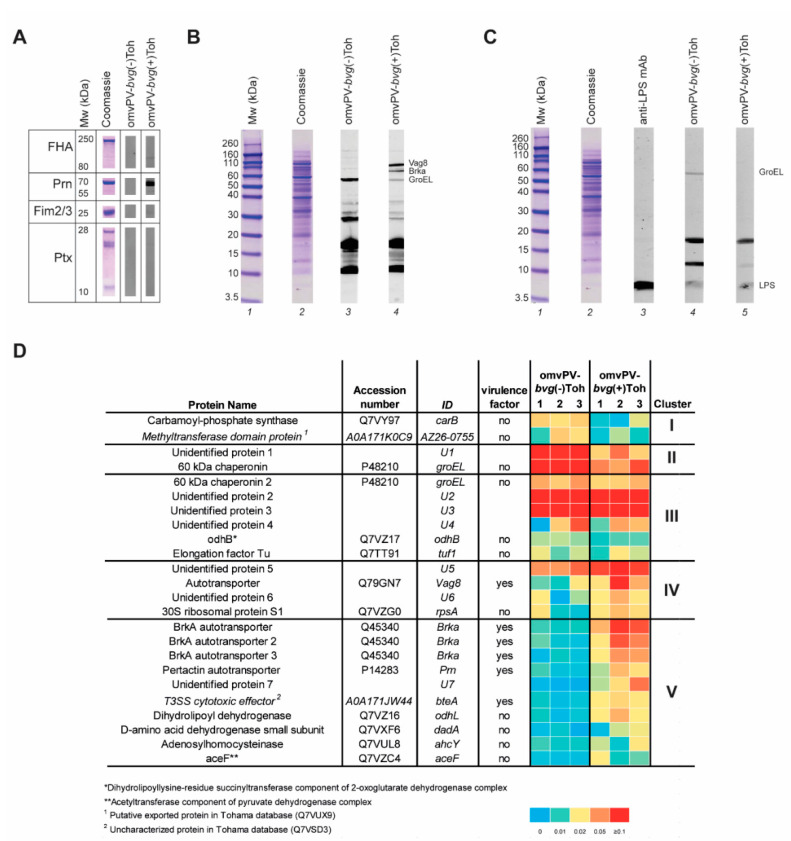
Immunoproteomic profiling of serum antibodies induced by omvPV-*bvg*(-)Toh and omvPV-*bvg*(+)Toh. (**A**) For IgG antibody profiling of pertussis antigens (FHA, Prn, Fim2/3 and Ptx, 1 µg), these purified proteins were individually separated on SDS-PAGE (proteins). Blots were incubated with pooled serum of mice immunized with omvPV-*bvg*(-)Toh or omvPV-*bvg*(+)Toh. (**B**) For total IgG antibody profiling, B1917 lysate (10 µg) was separated by SDS-PAGE and either Coomassie stained (lane 2) or blotted and incubated with pooled serum of mice immunized with omvPV-*bvg*(-)Toh (lane 3) and omvPV-*bvg*(+)Toh (lane 4). A protein marker is shown in lane 1. (**C**) For IgG3 antibody profiling, B1917 lysate (10 µg) was separated by SDS-PAGE and either Coomassie stained (lane 2) or blotted and incubated with anti-LPS monoclonal antibody (Control) (lane 3) or pooled serum of mice immunized with omvPV-*bvg*(-)Toh (lane 3) and omvPV-*bvg*(+)Toh (lane 4). A protein marker is shown in lane 1. All blots were visualized using IR-800 labeled goat-anti-mouse IgG or IgG3 antibodies. (**D**) Protein name, accession number, ID, virulence status, and spot intensity (average gray values) of 24 spots obtained from triplicate of total IgG 2DE blots incubated with pooled serum from mice immunized with either omvPV-*bvg*(-)Toh or omvPV-*bvg*(+)Toh (Appendix A). Spots are clustered depending on correlation in occurrence between both groups. Two filters were applied on the data for the clustering. (i) The average gray value of the triplicates had to be ≥0.01 and the signal should be present on two blots within a group. (ii) If the signal in both groups is ≥0.01, a fold change of ≥2 was applied to determine if the signal was increased in one group compared to the other group.

**Figure 5 vaccines-08-00429-f005:**
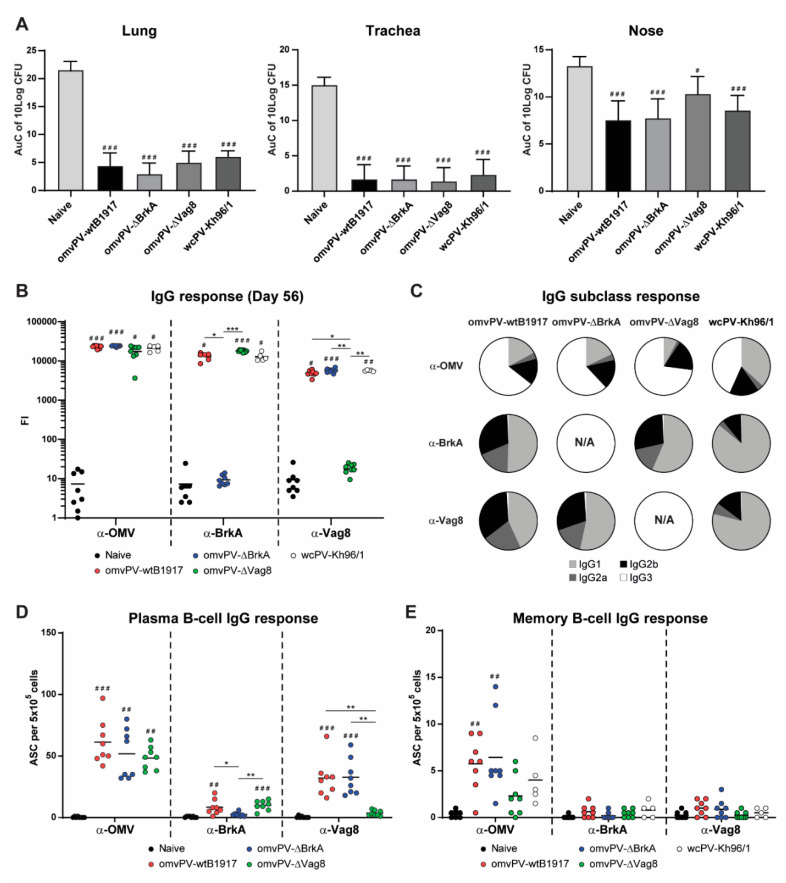
Level of protection and humoral responses of outer membrane vesicle pertussis vaccine (omvPV). (**A)** The level of protection in lungs, trachea and nose expressed as area under the curve (AuC) of 10log colony forming units (CFU) during 7 days after intranasal challenge of mice with virulent *B. pertussis* strain B1917. (**B**) Anti-OMV, anti-BrkA, anti-Vag8 IgG antibody responses and (**C**) underlying IgG subclass distribution in serum taken on day 56 of naive mice or mice vaccinated with omvPV-wtB1917, omvPV-∆BrkA, omvPV-∆Vag8 or wcPV-Kh96/1. Fluorescence intensity (FI) values are depicted on a ^10^log scale axis. (**D**) Number of IgG-producing plasma B-cells per 5 × 10^5^ splenocytes on day 35 in naive mice or mice vaccinated with omvPV-wtB1917, omvPV-∆BrkA, or omvPV-∆Vag8. (**E**) IgG memory B-cell responses per 5 × 10^5^ splenocytes on day 56 in naive mice or mice vaccinated with omvPV-wtB1917, omvPV-∆BrkA, omvPV-∆Vag8 or wcPV-Kh96/1. Significant differences compared to the naive mice are indicated as ^# ^*p* ≤ 0.05, ^##^
*p* ≤ 0.01 and ^###^
*p* ≤ 0.001 and actual *p*-values are shown in Appendix A. Significant differences between experimental groups are depicted with a line between both groups and * *p* ≤ 0.05, ** *p* ≤ 0.01 and *** *p* ≤ 0.001.

**Figure 6 vaccines-08-00429-f006:**
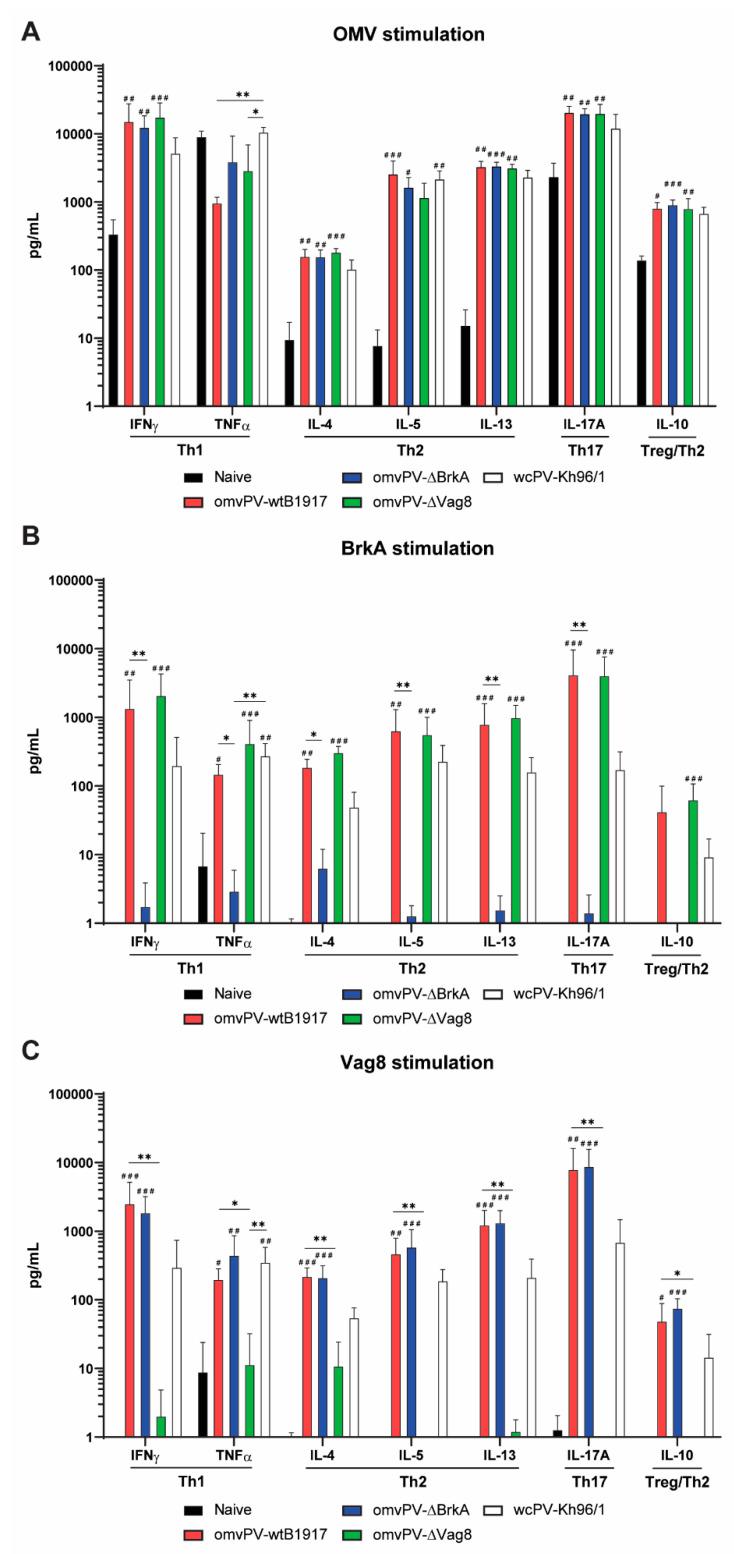
T-cell response. The concentrations of IFNγ, TNFγ (Th1-response), IL-4, IL-5, IL-10, IL-13 (Th2-response), and IL-17A (Th17-response) cytokines were determined in supernatants after restimulation of splenocytes with (**A**) OMV, (**B**) BrkA or (**C**) Vag8 of naive mice and mice vaccinated with omvPV-wtB1917, omvPV-ΔBrkA, omvPV-ΔVag8, or wcPV-Kh96/1. Concentrations (pg/mL) are depicted on a ^10^log scale axis. Significant differences compared to naive mice are indicated as ^#^
*p* ≤ 0.05, ^##^
*p* ≤ 0.01 and ^###^
*p* ≤ 0.001, and actual *p*-values are shown in Appendix A. Significant differences between experimental groups are depicted with a line between both groups and * *p* ≤ 0.05, ** *p* ≤ 0.01 and *** *p* ≤ 0.001.

**Table 1 vaccines-08-00429-t001:** List of virulent outer membrane proteins.

Accession Number	Name	ID
P12255	Filamentous hemagglutinin	fhaB
P14283	Pertactin autotransporter	Prn
P33410	Outer membrane usher protein FimC	fimC
P81549	Probable TonB-dependent receptor BfrD	bfrD
Q45340	BrkA autotransporter	BrkA
Q79GG1	Outer membrane porin protein OmpQ	ompQ
Q79GN7	Autotransporter	Vag8
Q79GQ6	Putative type III secretion protein	bscJ
Q79GR8	Putative type III secretion protein	bscC
Q79GX8	Tracheal colonization factor	TcfA
Q7VSD3	Type III secretion system cytotoxic effector	BteA
Q7VSG8	Putative heme receptor	hemC
Q7VVD6	Autotransporter	bapC
Q7VVJ2	Adhesin	fhaS
Q7VZ27	Putative outer membrane ligand binding protein	bipA
Q7VZP0	Probable TonB-dependent receptor for iron transport	bfrE

**Table 2 vaccines-08-00429-t002:** Primers used in this study.

Primer Name	Primer Sequence (5′→3′)	PCR
BrkA U-F	gcctcttcgccaaagaagg	Fusion Ia, fusion II
BrkA U-R2	GGAGCTCGCTCAGAAGCTGTgagaagttgaacaaaccgac	Fusion Ia
BrkA D-F2	GTCGGTTTGTTCAACTTCTCacagcttctgagcgagctcc	Fusion Ib
BrkA D-R	agaaggcgtggttcctgg	Fusion Ib, II
BrkA check-F2	ttcaggaaagctcttgttgg	Deletion check
BrkA check-R2	cggcatggacttctaagtcc	Deletion check
Vag8 U-F	ctgcgtcaaccgcttagc	Fusion Ia, fusion II
Vag8U-R2	GGTCACCAGCTGTAGCGATACctcaacacctcttggctag	Fusion Ia
Vag8D-F2	CTAGCCAAGAGGTGTTGAGgtatcgctacagctggtgacc	Fusion Ib
Vag8-R	aagccgcgtgcgactacgtc	Fusion Ib, II
Vag8 check-F2	ggcgttttctgtcaatcgtc	Deletion check
Vag8 check-R2	gccgaactgcaacgctactg	Deletion check

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
