# Peer review of "The Role of Virulence Proteins in Protection Conferred by Bordetella pertussis Outer Membrane Vesicle Vaccines"

_vaccines, 2020, doi:10.3390/vaccines8030429_

Round 1
Reviewer 1 Report
Major.
# The Introduction section looks over descriptive and some parts should be moved to discussion.
# As there are many treatments, authors are requested to provide the type of treatments in a tabular form and orders of the treatment in table should match with the figures.
# Authors are requested to cut short the result section. Earlier reports in the result section should be moved to discussion. Some of the result sections, authors started abruptly without explaining rational and they should put brief conclusion (1-2 line) in each section. (eg. Line no. 135).
# Authors should not mix up methods and results together. From the results, they should take out methods.
# In the result section- treatments effects should be explained in the order as type these were kept in the order of tabular form. Otherwise it makes lot of confusion as there are many treatments.
# Author must mention the intra-assay, inter-assay, and inter-operator variability for ELISPOT assay and Multiple Immunoassay used for antibody and/or cytokine measurement to ensure the reliability of each assay.
# Does author run the normality test before performing statistical analysis? If not, then they should perform it and apply to each statistical analysis.
# Author must use a unique scientific symbol in figures to describe the actual P values to avoid confusion and better understanding of the data.
# There are too many treatments and figure legends are not well described. Sometimes annotations of the figures are not consistent. (e.g. Supp fig. S2. A. Naïve was kept at the end while in B. it was kept in the middle. As a reader I would like to see annotations of the labelling are consistent throughout the figures.
# In the present study authors have enrolled 8 no. of animals in each experimental group, 5 no. of animals in reference group and again for colonization assay a total 5 no. of animals. Author must mention how the sample sizes were calculated in the present study.
# Figure 2. Should specify the color of the bar.
# Figure 5. Authors mentioned cytokines from T cells. How the T cells were purified and cultured?
Minor:
- Author must spell out the full form of abbreviations in the figure legends.
- Typographical errors in the text.
Author Response
We thank the reviewer for the constructive comments and suggestions. We have addressed these points and feel that the manuscript has improved significantly. Please note that the line and page numbers mentioned correspond to the manuscript with track changes.
Q1.1 # The Introduction section looks over descriptive and some parts should be moved to discussion.
A1.1 We have revised the introduction of our manuscript and have altered the following sections in order to make the introduction less descriptive. The section (Page 3, Line 64-67) was moved to discussion (Page 11, Line 279-281). In addition, the section (Page 3, Line 69-72) was altered.
Q1.2 # As there are many treatments, authors are requested to provide the type of treatments in a tabular form and orders of the treatment in table should match with the figures.
A1.2 We have added a Table (Table 1) to the manuscript to describe the different type of treatments and the isolation of samples for the different analysis of the two animal experiments. We hope that this provides a better overview of the study.
Q1.3 # Authors are requested to cut short the result section. Earlier reports in the result section should be moved to discussion. Some of the result sections, authors started abruptly without explaining rational and they should put brief conclusion (1-2 line) in each section. (eg. Line no. 135).
A1.3 We have shortened the results section by moving some parts to the material and methods section (Page 4, Line 89-99) and removing some references to earlier work such as (Page 8, Line 219-220). In addition we have added brief conclusions to each section such as:
“In conclusion, the level of virOMPs in omvPV is positively related to the degree of protection in the lungs against a B. pertussis infection.” (Page 5, Line 125-127)
“To conclude, deletion of either Vag8 or BrkA in omvPV had no effect on the level of bacterial colonization throughout the respiratory tract. This indicates that protection provided by the omvP-induced immunity is broader than only against either of its two most abundant immunogenic virulence factors.” (Page 7, Line 195-198)
“In conclusion, the deletion of either Vag8 or BrkA hardly had any effect on the magnitude, specificity and subclass distribution of omvPV-induced antibody responses, with the exception of responses against the deleted antigen itself.” (Page 8-9, Line 233-236)
“To conclude, the deletion of neither Vag8 nor BrkA affected the total numbers of antigen-specific IgG-producing plasma B-cells and IgG memory B-cells.” (Page 9, Line 252-253)
“To conclude, the mixed Th1/Th2/Th17 response induced by omvPV was not influenced by the deletion of either Vag8 or BrkA.” (Page 10, Line 272-273)
Q1.4 # Authors should not mix up methods and results together. From the results, they should take out methods.
A1.4 We have screened the results section and removed some sections that described methods. Sections we have changed are for instance: (Page 4, Line 89-99) (Page 5, Line 117-121) (Page 7, Line 183-185)
Q1.5 # In the result section- treatments effects should be explained in the order as type these were kept in the order of tabular form. Otherwise it makes lot of confusion as there are many treatments.
A1.5 We looked at the order of the description of the results and the way these are depicted in the figures. Overall, the order in text and figures is in line with one exception that we added on purposes in Figure 2A and B that we think increases the strength of this figure. In Figure 2A, the data of lung colonization of the second experiment (white bars) is already depicted whereas in text this is described after Figure 3. However, this order allowed us to make a correlation between %VirOMP and protection in the lungs for all vaccines tested in Figure 2B. If we would change the order, that would result in two separate figures whereas we think that Figure 2B is the axe of the complete manuscript. We hope that we can convince the reviewer with this background information and plea to keep the current order.
Q1.6 # Author must mention the intra-assay, inter-assay, and inter-operator variability for ELISPOT assay and Multiple Immunoassay used for antibody and/or cytokine measurement to ensure the reliability of each assay.
A1.6 For the analysis of both B-cell responses in the ELISpot and antibody responses in the MIA we make sure that all samples are run on the same plate. If it is not possible to measure all samples on one plate, these samples are randomized over multiple plates to limit potential biased intra-assay variance between groups. In addition, the inclusion of 8 animals per group demonstrates the variance within groups. Intra-assay variance is possible but limited and here we compare responses within an experiment by measuring and comparing all samples and controls at once under the same conditions. Moreover, for the antibody assay with the MIA, an institutional reference sample is included to monitor the reliability of the assay performance and to trend the results of all animal experiments.
Q1.7 # Does author run the normality test before performing statistical analysis? If not, then they should perform it and apply to each statistical analysis.
A1.7 The normality test is indeed part of our statistical analysis but was not mentioned specifically in the Material and Methods. We have added this information to clarify this method as follows:
“Differences in antibody levels, splenocyte cytokine levels, and B-cell numbers between groups were first tested for normality and subsequently analyzed using the Kruskal-Wallis test.” (Page 21, line 603-604)
Q1.8 # Author must use a unique scientific symbol in figures to describe the actual P values to avoid confusion and better understanding of the data.
A1.8 We agree with the reviewer on this point that with the current illustration of the statistical significance compared to the naïve group it is not possible to immediately see the magnitude of significance without looking at the Supplementary Tables. The most common method would be to depict all the significant differences of all comparisons with * depending on the p-value. However, the current figures all contain many comparisons, which would result in a complex depiction of many lines. We have tried this for a few figures when drafting the manuscript but this leads in our opinion to a reduced readability of the figures. As we agree with the reviewer that it is better to have an indication of statistical significance in one view for each figure we have reconsidered our strategy. We have decided to still depict the statistical difference between experimental groups and the naive groups with # but now vary in a range between # and # # # # depending on the p-value. By using the same strategy for the comparison between experimental groups but then using *, it is also possible in one sight to distinguish these comparisons. We hope we can convince the reviewer with this revised strategy which has increased the readability of all Figures in this manuscript.
Q1.9 # There are too many treatments and figure legends are not well described. Sometimes annotations of the figures are not consistent. (e.g. Supp fig. S2. A. Naïve was kept at the end while in B. it was kept in the middle. As a reader I would like to see annotations of the labelling are consistent throughout the figures.
A1.9 We have revised our figures and changed the order and labeling of groups to make it consistent. This has resulted in changes in Figures 2A, 2B, and Supp Figure S2A and B that can be found in the manuscript file. We have made some small changes to Figure legends such as Figure 3 to increase the readability. Finally, we have not changed the amount of treatments as these were all required to draw the conclusions in this manuscript. We hope that the new Table 1 we added following our answer to Q1.2 by the same reviewer will increase the readability of the manuscript.
Q1.10 # In the present study authors have enrolled 8 no. of animals in each experimental group, 5 no. of animals in reference group and again for colonization assay a total 5 no. of animals. Author must mention how the sample sizes were calculated in the present study.
A1.10 The total number of 5 mice for the assessment of lung colonization is based on a power calculation and gathered datasets from previous experiments. With this number of mice we calculated that we could determine a difference of 0.9 log10 or more with a power of 80%. For the analysis of immunological parameters such as B-cell and T-cell responses we have experienced in the past that 8 mice per group allow a reliable analysis taking into account variation and occasional loss of sample. The reason why the wcPV reference group had 5 mice instead of 8 mice is because this group was first only included to compare level of colonization and antibody response to the omvPV groups. However, it was later decided still to include the animals sacrificed on day 56 for analysis of B-cells and T-cells to also compare these parameters to those induced by omvPV. Unfortunately, the number of animals could not be changed. This comparison is therefore still possible but with a lower power. To clarify the number of animals we have added the following to the materials and methods section:
“The total number of mice for the assessment of lung colonization (n = 5) is based on a power calculation (difference ≥ 0.9 log10, power = 80%) using datasets from previous experiments. For the analysis of immunological parameters such as B-cell and T-cell responses, 8 mice per group allowed a reliable analysis.” (Page 21, line 593-596)
Q1.11 # Figure 2. Should specify the color of the bar.
A1.11 The black and white illustration of the bars was to distinguish between experiment 1 and 2. However, this explanation was indeed not specified in the legend. There, we have added this.
“The level of protection of all vaccines under test in this study, divided over experiment 1 (black bars) and 2 (white bars), expressed as AuC of 10log lung colony forming units (CFU) during 7 days after intranasal challenge of mice with virulent B. pertussis strain B1917.” (Page 25, line 641-643)
Q1.12 # Figure 5. Authors mentioned cytokines from T cells. How the T cells were purified and cultured?
A1.12 For the analysis of vaccine-induced T-cell responses we have used a common method that was described by our group and others (Raeven et al. Mucosal Imm 2018, Brummelman et al. 2015). In this method we stimulate isolated splenocytes with antigens (OMV, Vag8 and BrkA) for three days and subsequently determine the concentration of T-cell related cytokines in the culture supernatant. It is important to note that in this method the T-cells are not purified or cultured separately. This common method has been recognized to determine T-cell responses. However this method is not as detailed as a Flow cytometric analysis that was not available at the time of performing these experiments.
Minor:
Q1.13 Author must spell out the full form of abbreviations in the figure legends.
A1.13 We have spelled out abbreviations where applicable in the figure legends.
Q1.14 Typographical errors in the text.
A1.14 We have screened the manuscript and have changed some typographical errors.
Reviewer 2 Report
This is a magnum opus of a study, set out to present murine protection studies exploring the properties of experimental OMV vaccine candidates to address the very real problem of the need to revisit the nature of improved B. pertussis vaccines in the face of concern over cases and limited protection. The manuscript is really comprehensive and makes a number of really informative points.
The studies are carefully done and nicely reported, right down to proteomic appraisal of the preparations. All in all, this was a joy to read.
I have a few minor comments:
- I probably missed this, but I could not spot where the strain, sex and age of the mice were described?
- It was perhaps a little disappointing that, while B cell subsets were to some extent described, CD4 subset responses were just inferred by reading off cytokine levels in splenocyte supernatants. Possibly outside the scope of this study, but they might consider looking in far more detail at the nature of T cell memory subsets and effector functions associated with protection sometime in the future.
Author Response
We thank the reviewer for the constructive comments and suggestions. We have addressed these points that improved the readability of the manuscript. Please note that the line and page numbers mentioned correspond to the manuscript with track changes.
Q2.1 I probably missed this, but I could not spot where the strain, sex and age of the mice were described?
A2.1 This information was indeed lacking. Therefore, we have added this as follows:
“Here, female, 8-week old BALB/c mice (Harlan, The Netherlands) were vaccinated twice (day 0 and day 28) with either…” (Page 18, Line 495).
Q2.2 It was perhaps a little disappointing that, while B cell subsets were to some extent described, CD4 subset responses were just inferred by reading off cytokine levels in splenocyte supernatants. Possibly outside the scope of this study, but they might consider looking in far more detail at the nature of T cell memory subsets and effector functions associated with protection sometime in the future.
A2.2 We agree with the reviewer that the current assessment of T-cell responses is ‘only’ an indirect analysis based on cytokine production in supernatants. At the time of performing these studies this was the only available assessment and in addition, we indeed also decided that based on the scope of the study, this data was sufficient to answer the current research question. In the future we aim to investigate the T-cell responses in more depth again.
Reviewer 3 Report
In this article, the authors test the hypothesis that the virulence protein composition of outer membrane vesicle vaccines against Bordetella pertussis changes their immunogenic protection and response. Using a wide variety of analysis techniques from LC-MS to in vivo mouse assays to 2D SDS-PAGE/Western Blot to genetic manipulations, the authors test the protective effects of virulence proteins in outer membrane vesicle vaccines, the immunogenic responses elicited by different vaccines, and the necessity for specific virulence proteins in vaccination efficiency. The statistical analysis is rigorous and transparent and the level of molecular characterization is thorough. This article will be interesting to the readership of Vaccines and is ready for publication with only minor modifications.
Revisions:
- Lines 390-391: Claims about individual virulence protein as not essential to the immunogenic response seems premature as only two virulence genes (BrkA and Vag8) were tested individually but over 10 virulence genes were found in only the +bvg strain. This statement should be qualified to better reflect their data.
- Figure 1B: Statistical markers should be added to the total virulence OMPs in 1B.
- Figure 3: How were the five subgroups assigned? What statistical threshold or method for clustering was used? This information should be added to figure legend.
- Figures 2 and 4: Statistical analysis is given in Tables S2-S4 but it is difficult to correlate this table data with the information in Figures 2 and 4. The authors should try to add more designations besides just # to differentiate the statistical levels on the figures. They can still include Supplemental Tables S2-S4 as evidence for the total statistical rigor.
Author Response
We thank the reviewer for the constructive comments and suggestions. We have addressed these points and feel that the manuscript has improved significantly. Please note that the line and page numbers mentioned correspond to the manuscript with track changes.
Q3.1 Lines 390-391: Claims about individual virulence protein as not essential to the immunogenic response seems premature as only two virulence genes (BrkA and Vag8) were tested individually but over 10 virulence genes were found in only the +bvg strain. This statement should be qualified to better reflect their data.
A3.1 We agree with the reviewer that the current statement is perhaps too strong as we have not investigated the role of all VirOMP found in our omvPV individually. Both Vag8 and BrkA are by far the most abundant VirOMP we detected in our omvPV that is also reflected in the immune response observed against these antigens. However, we acknowledge that also immunity directed against less abundant VirOMP such as Prn can have a major role in the protection provided by omvPV. We have changed our statement as follows:
“However, our data demonstrated that although the strength of omvPV induced immunity relies on the presence of virOMPs, this is not solely depending on either of the two most abundant immunogenic antigens (BrkA or Vag8).” (Page 13, Line 360-362)
Q3.2 Figure 1B: Statistical markers should be added to the total virulence OMPs in 1B.
A3.2 The total sum of virulence OMPs as depicted in Figure 1B is obtained for single analysis with the LC-MS on the products or from technical replica’s. Therefore, it is unfortunately not possible to add error bars or calculate statistical differences between the products in this figure. We have used this analysis to characterize the products that were used later on in the animal experiments, so therefore no multiple batches were produced. Based on other data generated in our lab regarding this method has shown that the CV of virulence factor content is approximately 6%.
Q3.3 Figure 3: How were the five subgroups assigned? What statistical threshold or method for clustering was used? This information should be added to figure legend.
A3.3 The five clusters were assigned based on a strategy where two filters were applied on the data for the clustering. i) The average grey value of the triplicates had to be ≥ 0.01 and the signal should be present on 2 blots within a group. ii) If the signal in both groups is ≥ 0.01 a fold change of ≥ 2 was applied to determine if the signal was increased in one group compared to the other group. This strategy was already described in the materials and methods section (Page 21, Line 587-590) but we have also added it to the legend of Figure 3 (Page 28, Line 662-665) to increase the readability of this figure.
Q3.4 Figures 2 and 4: Statistical analysis is given in Tables S2-S4 but it is difficult to correlate this table data with the information in Figures 2 and 4. The authors should try to add more designations besides just # to differentiate the statistical levels on the figures. They can still include Supplemental Tables S2-S4 as evidence for the total statistical rigor.
A3.4 We agree with the reviewer on this point that with the current illustration of the statistical significance compared to the naïve group it is not possible to immediately see the magnitude of significance without looking at the Supplementary Tables. The most common method would be to depict all the significant differences of all comparisons with * depending on the p-value. However, the current figures all contain many comparisons, which would result in a complex depiction of many lines. We have tried this for a few figures when drafting the manuscript but this leads in our opinion to a reduced readability of the figures. As we agree with the reviewer that it is better to have an indication of statistical significance in one view for each figure we have reconsidered our strategy. We have decided to still depict the statistical difference between experimental groups and the naive groups with # but now vary in a range between # and # # # # depending on the p-value. By using the same strategy for the comparison between experimental groups but then using *, it is also possible in one sight to distinguish these comparisons. We hope we can convince the reviewer with this revised strategy which has increased the readability of all Figures in this manuscript.
Round 2
Reviewer 1 Report
# Table 1 looks like incomplete. What do they mean by Experimental design? The complete header is missing. Need to expand someway the types of treatment so that it goes a complete message to the reader.
# Line 84. What do authors mean by high to low? The sentence looks like redundant.
# Authors were suggested to conclude each sub-headings of the result in the first round of review. Please remove the word “conclusion”. It is suggested to reconstruct the sentences with- “these results provide support”, “this observation suggested” etc. They should conclude the whole study at the end of the discussion.
# Line no. 616-617. Why only Kruskal-Wallis? None of the study groups was normally distributed?
# scientific symbols in figures * and # looks like the same and makes it very confusing. They should keep the statistical level up to p < 0.001.
# Level of cytokines if they are in the log scale, it should appear in the figure.
# Still appears typographical errors in the text.
Author Response
We thank the reviewer for these additional comments and suggestions. We have addressed these points and that further improved the manuscript. Please note that the line and page numbers mentioned correspond to the manuscript with track changes.
Comments and Suggestions for Authors
Q1.1 # Table 1 looks like incomplete. What do they mean by Experimental design? The complete header is missing. Need to expand someway the types of treatment so that it goes a complete message to the reader.
A1.1 As the included Table in the previous revised version of the manuscript was not self-explanatory for the reviewer, and therefore the reader, we decided to depict the experimental design of both experiments in this study in a Figure (New Figure 1 in revised manuscript). In this Figure we have included more information on the experimental design and type of treatments as well as the performed analysis described in this study including references to the other Figures where this data is depicted. We think that this way of illustrating the experimental design allows a better readability and understanding of the current study for the readers.
Q1.2 # Line 84. What do authors mean by high to low? The sentence looks like redundant.
A1.2 With “high to low” we referred to the range of percentage of VirOMP in the three omvPV products that we produced. We think that this is important information for a reader to understand the order of abundance of virOMP in the included omvPV. As this was not clear in the previous revised manuscript, we have changed the sentence to clarify this issue as follows:
“To investigate the role of virOMP, a series of omvPV varying in percentage of virOMP (ranging from high to low % VirOMP: omvPV-wtB1917, omvPV-bvg(+)Toh and omvPV-bvg(-)Toh) was produced…” (page 4, line 84-86)
Q1.3 # Authors were suggested to conclude each sub-headings of the result in the first round of review. Please remove the word “conclusion”. It is suggested to reconstruct the sentences with- “these results provide support”, “this observation suggested” etc. They should conclude the whole study at the end of the discussion.
A1.3 We have added conclusions to the sub-headers in the results and changed all concluding sentences as follows:
Sub-header 1: The level of virOMP in omvPV is positively related to protection against B. pertussis.
“Overall, this observation suggests that the level of virOMPs in omvPV is positively related to the degree of protection in the lungs against a B. pertussis infection.” (Page 5, Line 111-113)
Sub-header 2: Immunoproteomic profiling of high antibody responses induced by omvPV-bvg(-)Toh and omvPV-bvg(+)Toh reveals partial distinct antigen specificity.
“Overall, these data reveal that both omvPV-bvg(+)Toh and omvPV-bvg(-)Toh induce strong antibody responses in mice that show some overlap in antibody profiles, yet particularly the ones against virulence factors were distinct in presence or magnitude, which may explain the difference in level of protection between both vaccines.” (Page 6, Line 151-154)
Sub-header 3: Deletion of either Vag8 or BrkA in outer membrane vesicles does not affect its protective capacity and immunity profiles
Sub-header 4: Immunization with omvPV-∆BrkA and omvPV-∆Vag8 reveals equal protective capacity as omvPV-wtB1917
“These results provide support that protection provided by the omvP-induced immunity is broader than only against either of its two most abundant immunogenic virulence factors as deletion of either Vag8 or BrkA in omvPV had no effect on the level of bacterial colonization throughout the respiratory tract.” (Page 7, Line 184-187)
Sub-header 5: Deletion of BrkA or Vag8 has limited effect on magnitude, specificity and subclass distribution of antibody responses against other antigens
These observations indicate that the deletion of either Vag8 or BrkA hardly had any effect on the magnitude, specificity and subclass distribution of omvPV-induced antibody responses, with the exception of responses against the deleted antigen itself.” (Page 8, Line 223-225)
Sub-header 6: Number of omvPV-induced plasma and memory B-cells are not influenced by deletion of BrkA or Vag8
“Therefore, these observations support that the deletion of neither Vag8 nor BrkA affected the total numbers of antigen-specific IgG-producing plasma B-cells and IgG memory B-cells induced by omvPV.” (Page 9, Line 241-243)
Sub-header 7: The omvPV-induced mixed Th1/Th2/Th17 response is not affected by the deletion of either Vag8 or BrkA
“These observations indicate that the mixed Th1/Th2/Th17 response induced by omvPV was not influenced by the deletion of either Vag8 or BrkA.” (Page 10, Line 263-264)
Q1.4 # Line no. 616-617. Why only Kruskal-Wallis? None of the study groups was normally distributed?
A1.1 We first tested normal distribution. As the majority of groups was not normally distributed we performed a non-parametric test (like the Kruskal-Wallis) on all groups instead of using different tests within an analysis depending on the normal distribution.
Q1.5 # scientific symbols in figures * and # looks like the same and makes it very confusing. They should keep the statistical level up to p < 0.001.
A1.5 The current depicted strategy was meant to allow a reader to distinguish the comparison between experimental groups and the saline group (#) from the comparison of experimental groups among each other (* with line). To extra clarify this, we did not change the symbols (* and #), since we think there is no better alternative. However, we have now explained in the Figure legend that this line is only present for the latter comparison (between experimental groups). Moreover, we agree with the reviewer that limiting the statistical comparison only up to p ≤ 0.001 will improve the readability of the figures so we have changed this in all Figures, legends and the Methods section.
Q1.6 # Level of cytokines if they are in the log scale, it should appear in the figure.
A1.6 The axis of the cytokines and the antibody levels are set on Log scale, and actual values (and not log transformed values) are depicted. We have changed the figure legends to explain this..
Q1.7 # Still appears typographical errors in the text
A1.7 We have screened the manuscript and used the automatic spell check and detected the following typographical errors. If we, despite the effort, have overseen any other errors that the reviewer has observed it would be helpful if the reviewer could notify the exact location.
- In line 141 and in Figure 4 we changed methylatransferase into methyltransferase
- We changed line 140-141 as follows: “…directed against two cytosolic proteins, carB and the methyltransferase domain protein”
- In Figure legend 5, 6, S3 and S4 we have changed 10 Log into 10
- In line 185 we have changed “omvP-induced immunity” into “omvPV-induced immunity”
- In line 237-238 we have changed the following: “…as compared to naive mice but there was only a significant difference was seen for groups immunized with…”
- In line 295-298 we have changed the following: “Culturing pertussis strain B1917 cultured in THIJS medium results in presence of a high percentage of OMP in omvPV that are almost all virOMPs. Culturing Tohama I , cultured in a non-defined Verwey medium also resulted in high expression of virOMPs in on omvPVs,…”
- In line 333 we have changed the following: “…against Prn-expressing and Prn-deficient…”
- We added a space between (Promega). and Solid-phase in line 431
- A space was added between % and VirOMP in line 634
- We added spaces between 260 – 3 kDa in Figure S1
- In supplemental table S1 the value for Prn sample of omvPV-bvg(-)Toh 3 was mistyped and changed from 00.432 into 0.0043
Round 3
Reviewer 1 Report
The authors sincerely corrected all the comments raised.